# REFINING THE VARIATIONAL POSTERIOR THROUGH ITERATIVE OPTIMIZATION

## ABSTRACT

Variational inference (VI) is a popular approach for approximate Bayesian inference that is particularly promising for highly parameterized models such as deep neural networks. A key challenge of variational inference is to approximate the posterior over model parameters with a distribution that is simpler and tractable yet sufficiently expressive. In this work, we propose a method for training highly flexible variational distributions by starting with a coarse approximation and iteratively refining it. Each refinement step makes cheap, local adjustments and only requires optimization of simple variational families. We demonstrate theoretically that our method always improves a bound on the approximation (the Evidence Lower BOund) and observe this empirically across a variety of benchmark tasks. In experiments, our method consistently outperforms recent variational inference methods for deep learning in terms of log-likelihood and the ELBO. We see that the gains are further amplified on larger scale models, significantly outperforming standard VI and deep ensembles on residual networks on CIFAR10.

## 1 INTRODUCTION

Uncertainty plays a crucial role in a multitude of machine learning applications, ranging from weather prediction to drug discovery. Poor predictive uncertainty risks potentially poor outcomes, especially in domains such as medical diagnosis or autonomous vehicles where some forms of high confidence errors may be especially costly (Amodei et al., 2016). Thus, it is becoming increasingly important that the underlying model provides high quality uncertainty estimates along with its predictions. Yet, possibly the most widely used models, deep neural networks (LeCun et al., 2015), are unable to accurately quantify model uncertainty. They are often overconfident in their predictions, even when their predictions are incorrect (Guo et al., 2017; Ovadia et al., 2019).

By marginalizing over a posterior distribution over the parameters given the training data, Bayesian inference provides a principled approach to capturing uncertainty. In contrast, standard training of neural networks employs a point estimate of the parameters, which cannot account for model uncertainty. Unfortunately, exact Bayesian inference is intractable in general for neural networks. To model epistemic uncertainty, variational inference (VI) instead approximates the true posterior with a simpler distribution. The most widely used one for neural networks is the mean-field approximation, where the posterior is represented using an independent Gaussian distribution over all the weights. Variational inference is appealing since it reduces the problem of inference to an optimization problem, minimizing the discrepancy between the true posterior and the variational posterior. The key challenge, however, is the task of training expressive posterior approximations that can capture the true posterior without significantly increasing the computational costs.

This paper describes a novel method for training highly flexible posterior approximations. The idea is to start with a coarse, mean-field approximation $q(w)$ and make iterative, local refinements to it. The regions of the local refinements are determined by sampling the values of additive auxiliary variables. The model parameters $w$ are expressed using a number of auxiliary variables $a_k$ (Figure 1 left) for $k = 1, \ldots, K$ that leave the marginal distribution unchanged. In each iteration, we sample the value of an auxiliary variable according to the current variational approximation $q(a_k)$ and refine the approximation by conditioning on the newly sampled value $q(w) \approx p(w|x, y, a_{1:k})$ (Figure 1 right illustrates the process). Each refinement step makes cheap, local adjustments to the variational posterior in the region of the sampled auxiliary variables. At the end, we draw one sample from

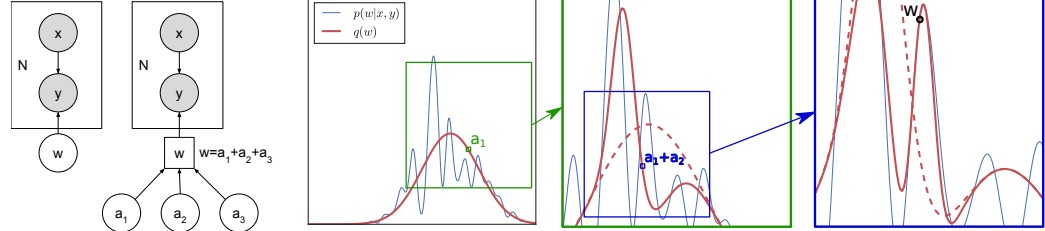

Figure 1: (Left) The supervised learning model and augmented model respectively where $w$ is expressed as a sum of independent auxiliary variables. (Right) Illustration of the refining algorithm. In each iteration the value of an auxiliary variable is fixed and the posterior is locally adjusted. In the final iteration, a sample is drawn from $w$. Through the iterations, the variational distribution is able to approximate well the true posterior in a small region.

the refined $q(w)$. The refinement iterations have to be repeated for each posterior sample. The algorithm results in samples from a highly complex distribution, starting from a simple mean-field approximation. While the distribution of the samples is difficult to quantify, we show that it is not limited to factorized, uni-modal forms, and that the procedure is guaranteed to improve the resulting ELBO without posing a significant computational overhead.

Summary of contributions:

- A novel algorithm for refining a variational distribution, increasing its flexibility.
- We show that the refinement steps are guaranteed to improve the quality of the variational distribution under mild conditions.
- We showcase the effectiveness of the method on Bayesian neural networks using a set of UCI regression and image classification benchmarks. We set a new state-of-the-art in uncertainty estimation using variational inference at ResNet scale (ResNet-20, (He et al., 2016)) scale on CIFAR10.

## 2 METHODS

In this section, we first describe standard variational inference (VI), followed by a detailed description of the iterative refinement algorithm. While VI and our proposed methodology are generally applicable to latent variable models, in this work, we consider the application to Bayesian neural networks (Figure 1), where $x$ are inputs, $y$ outputs, and $w$ the weights of network.

### 2.1 VARIATIONAL INFERENCE IN BAYESIAN NEURAL NETWORKS

Exact Bayesian inference in Bayesian neural networks (BNN) (Figure 1) is often intractable and is NP-hard in the general case. Variational inference attempts to approximate the true posterior $p(w|x, y)$ with an approximate posterior $q_\phi(w)$, typically from a simple family of distributions, for example independent Gaussians over the weights, i.e. the mean-field approximation. To ensure that the approximate posterior is close to the true posterior, the parameters of $q_\phi(w)$, $\phi$ are optimized to minimize their Kullback-Leibler divergence: $D_{\mathrm{KL}}(q_\phi(w) \parallel p(w|x, y))$. At the limit of $D_{\mathrm{KL}}(q_\phi(w) \parallel p(w|x, y)) = 0$, the approximate posterior exactly captures the true posterior, although this might not be achievable if $p(w|x, y)$ is outside of the distribution family of $q_\phi(w)$.

In order to minimize the KL-divergence, variational inference optimizes the Evidence Lower Bound (ELBO) w.r.t. $\phi$ (denoted as $\mathcal{L}(\phi)$), which is a lower bound to the log marginal likelihood $\log p(y|x)$. Since the the marginal likelihood can be expressed as the sum of the KL-divergence and the ELBO, maximizing the ELBO is equivalent to minimizing the KL divergence:

$$\log p(y|x) = \underbrace{D_{\mathrm{KL}}\big(q_\phi(w) \parallel p(w|x, y)\big)}_{\geq 0} + \mathcal{L}(\phi) \geq \mathcal{L}(\phi) = \mathbb{E}_{q_\phi}\big[\log p(y|x, w)\big] - D_{\mathrm{KL}}\big(q_\phi(w) \parallel p(w)\big),$$

due to non-negativity of the KL-divergence.

```
 1: procedure REFINEANDSAMPLE(φ)
 2:     for m = 1, . . . , M do
 3:         φ₀ ← φ
 4:         for k = 1, . . . , K do
 5:             a_k ~ q_{φ_{k-1}}(a_k)          ▷ Sample a_k
 6:             q_{φ_k}(w) ← q_{φ_{k-1}}(w|a_k)  ▷ Initialize φ_k
 7:             φ_k ← arg max_{φ_k} L_{|a_{1:k}}(φ_k)  ▷ Optimize φ_k
 8:         end for
 9:         w_m ~ q_{φ_K}(w)          ▷ Sample the refined posterior
10:     end for
11:     return w_{1:M}
12: end procedure
```

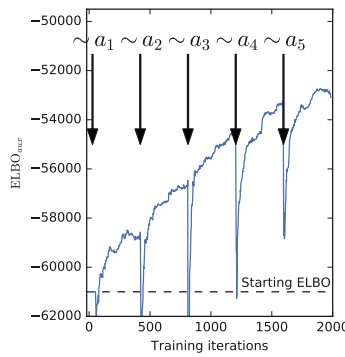

Algorithm 1: Pseudocode for the iterative refinement of the variational posterior

Figure 2: The ELBO staircase: $\text{ELBO}_{\text{aux}}$ is increasing as we sample the auxiliary variables (single sample Monte Carlo estimate, LeNet-5/CIFAR10).[1]

Following the optimization of $\phi$, the model can be used to make predictions on unseen data. For a new input $x'$, the predictive distribution $p(y'|x', y, x)$ can be approximated by stochastically drawing a small number (around $M \leq 10$) of sample model parameters and averaging their prediction in an ensemble model:

$$w_{1:M} \sim q_\phi(w), \quad p(y'|x', y, x) \approx \frac{1}{M} \sum_{i=1}^{M} p(y'|x', w_i).$$

## 2.2 REFINING THE POSTERIOR

The main issue with variational inference is the inflexibility of the posterior approximation. The most widely used variant of variational inference, mean-field variational inference, approximates the posterior with independent Gaussians across all dimensions. This approximation is too simplistic to capture the complexities of the posterior for complicated models such as BNNs. Our idea is to refine the posterior approximation through iterative optimization. Using the refinement procedure, it is feasible to train a detailed posterior in the regions of the posterior samples used for prediction while relying on a coarse-grained approximation further away from these samples.

The graphical model is augmented with a finite number of auxiliary variables $a_{1:K}$ as shown in Figure 1. The constraints are that $(x, y)$ must be conditionally independent of the auxiliary variables given $w$ and that they must not affect the prior distribution $p(w)$. This is important in justifying the use of the initial variational approximation. A significant way in which we distinguish ourselves from hierarchical variational models (Ranganath et al., 2016) is that the *model is unaffected by the presence of the auxiliary variables*. Their purpose is solely to aid the inference procedure. Given a Gaussian prior $\mathcal{N}(0, \sigma_w^2)$ over $w$, we express $w$ as a sum of independent auxiliary variables[2]

$$w = \sum_{k=1}^{K} a_k, \quad \text{with} \quad p(a_k) = \mathcal{N}(0, \sigma_{a_k}^2) \quad \text{for } k = 1, \dots, K,$$

while ensuring that $\sum_{k=1}^{K} \sigma_{a_k}^2 = \sigma_w^2$ so that the prior $p(w) = \mathcal{N}(0, \sigma_w^2)$ is unchanged.

Locally refining the approximate posterior refers to iteratively sampling the values of the auxiliary variables $a_{1:K}$ and then approximating the posterior conditional on the sampled values, i.e. $q_{\phi_k}(w)$ approximates $p(w|x, y, a_{1:k})$ for iterations $k = 1, \dots, K$ (Algorithm 1). Starting from the initial mean field approximation $q_\phi(w)$, we sample the value of $a_1$ from $q_\phi(a_1) = \int p(a_1|w)q_\phi(w)\, dw$,

---

[1]The sudden drops after sampling are optimizer artefacts due to having to reset the parameters of Adam.

[2]While we are focusing on one specific definition of the auxiliary variables, additive auxiliary variables, note that all of our results straight-forwardly generalize to arbitrary joint distributions $p(w, a_{1:K})$ that meet the constraints.

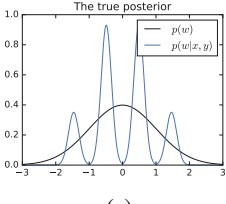 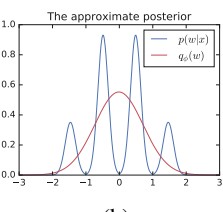 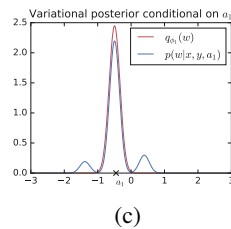 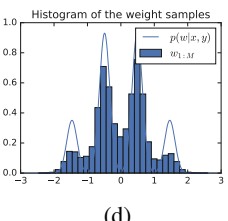

|(a)|(b)|(c)|(d)|

Figure 3: A simple multi-modal example demonstrating how our method can capture a more complex distribution by refining a simple mean-field posterior. In (a) the true posterior is too complex to be well approximated by a Gaussian. (b) The Gaussian approximate posterior after optimizing the ELBO (ELBO $= -1.79$). (c) After sampling $a_1$, we optimize the conditional ELBO w.r.t. $\phi_1|a_1$. $w_m$ is drawn from $q_{\phi_1|a_1}(w)$. (d) Samples from the refined posterior approximation. ELBO $\geq -1.45$.

then optimize the approximation to the conditional posterior: $q_{\phi_1}(w) \approx p(w|x, y, a_1)$. This procedure is then iteratively repeated for $a_{2:K}$. In iteration $k$,

$$\textbf{1)}\quad a_k \sim \int p(a_k|a_{1:k-1}, w)q_{\phi_{k-1}}(w)\,\mathrm{d}w \quad \textbf{2)}\quad \phi_k = \arg\min D_{\mathrm{KL}}\big(q_{\phi_k}(w) \,\big\|\, p(w|x, y, a_{1:k})\big).$$

Analogously to variational inference, the KL divergence is minimized through the optimization of the conditional ELBO in each iteration: $\mathcal{L}_{|a_{1:k}}(\phi_k) = \mathbb{E}_{q_{\phi_k}}\big[\log p(y|x, w)\big] - D_{\mathrm{KL}}(q_{\phi_k}(w) \,\|\, p(w|a_{1:k}))$. In order to get independent samples from the variational posterior, we have to repeat the iterative refinement for each ensemble member $w_{1:M}$.

**Toy example** We use a toy example to demonstrate the procedure. In this toy example, we have a single weight $w$ with prior $p(w) = \mathcal{N}(0, 1)$ and a complicated posterior with four modes (synthetically generated data). We express $w$ as the sum of two auxiliary variables $w = a_1 + a_2$ with $p(a_1) = \mathcal{N}(0, 0.8)$ and $p(a_2) = \mathcal{N}(0, 0.2)$ (which recovers $p(w)$ as per the constraint).

As Figure 3b shows, a Gaussian approximation to the posterior fails to capture the multimodal nature of the true posterior. The first step of the refinement is to sample using $q_\phi$: $a_1 \sim q_\phi(a_1) = \int p(a_1|w)q_\phi(w)\,\mathrm{d}w$. Next, we condition on the value of $a_1$ and initialize the parameters of the variational posterior accordingly: $q_{\phi_1}(w) = \mathcal{N}\big(w|\mu_{\phi_1}, \sigma_{\phi_1}^2\big) \leftarrow q_\phi(w|a_1)$. Both $q_\phi(a_1)$ and $q_\phi(w|a_1)$ can be computed analytically for Gaussian distributions. After optimizing $\phi_1$, the approximate posterior $q_{\phi_1}(w)$ is able to capture a good, local approximation to the posterior $p(w|a_1, x, y)$ (Figure 3c). In Figure 3d, we can see the histogram of the refined posterior, that is, the distribution we are generating samples from (for each sample from $w$, we drew a sample from $q_\phi(a_1)$ and optimized $q_{\phi_1}(w)$). Clearly, it is a much better approximation to the true posterior than the Gaussian approximation we started with, although it is important to note that the true posterior is not recovered exactly.

## 2.3 THEORETICAL JUSTIFICATION

Our theoretical claims are twofold. Firstly, that through this procedure, we are optimizing a lower bound to the ELBO and secondly, that the refinement cannot result in a worse posterior approximation than the initial mean-field approximation that we start with (in the ELBO sense). That is,

$$\mathrm{ELBO}_{\mathrm{ref}} \geq \mathrm{ELBO}_{\mathrm{aux}} \geq \mathrm{ELBO}_{\mathrm{init}}\,,$$

where $\mathrm{ELBO}_{\mathrm{ref}}$ denotes the ELBO of the refined posterior $q_{\mathrm{ref}}$, $\mathrm{ELBO}_{\mathrm{aux}}$ refers to the objective that the refinement process is optimizing, and $\mathrm{ELBO}_{\mathrm{init}}$ is the ELBO of the initial mean-field approximation.

**Lower bound to the ELBO** Consider the case with two auxiliary variables $a_1$ and $a_2$. The initial training optimizes the $\mathrm{ELBO}_{\mathrm{init}} = \mathbb{E}_{q_\phi}\big[\log p(y|x, w)\big] - D_{\mathrm{KL}}\big(q_\phi(w) \,\big\|\, p(w)\big)$ and the refinement step optimizes the conditional ELBO, $\mathcal{L}_{|a_1}(\phi_1) = \mathbb{E}_{q_{\phi_1}}\big[\log p(y|x, w)\big] - D_{\mathrm{KL}}\big(q_{\phi_1}(w) \,\big\|\, p(w|a_1)\big)$.

The key observation is that we can define $\mathrm{ELBO}_{\mathrm{aux}}$ that is a lower bound to $\mathrm{ELBO}_{\mathrm{ref}}$ and is increased both by the initial training and the refinement steps:

$$
\begin{aligned}
\mathrm{ELBO}_{\mathrm{aux}} &= \mathbb{E}_{q_\phi}\Big[\mathbb{E}_{q_{\phi_1}}\Big[\log p(y|x,w) - \log\frac{q_{\phi_1}(w)}{p(w|a_1)}\Big] - \log\frac{q_\phi(a_1)}{p(a_1)}\Big]\\
&= \mathbb{E}_{q_{\mathrm{ref}}}\big[\log p(y|x,w)\big] - D_{\mathrm{KL}}\big(q_{\mathrm{ref}}(w,a_1) \,\big\|\, p(w,a_1)\big)\\
&\leq \mathbb{E}_{q_{\mathrm{ref}}}\big[\log p(y|x,w)\big] - D_{\mathrm{KL}}\big(q_{\mathrm{ref}}(w) \,\big\|\, p(w)\big) = \mathrm{ELBO}_{\mathrm{ref}}\,,
\end{aligned}
$$

since the KL divergence of the joint distribution is greater than or equal to that of the marginals.

**Guarantee of improvement**  Improvement in the ELBO ($\mathrm{ELBO}_{\mathrm{ref}} \geq \mathrm{ELBO}_{\mathrm{init}}$) is guaranteed under two assumptions. First, that the conditional variational posterior, $q_\phi(w|a_1)$, is within the variational family of $q_{\phi_1}$. Second, that the process that optimizes $\phi_1$ does not make it worse than the value it was initialized with. The first assumption holds for Gaussian families: $q_\phi(w|a_1)$ is Gaussian and can be computed in closed form. The second assumption is reasonable to assume for most optimizers and, in addition, it can be ensured by comparing the initial value to the final value and choosing the one with the more desirable objective value.

The argument goes as follows. By initializing $q_{\phi_1}$ such that it coincides with $q_\phi(w|a_1)$, we can ensure that $\mathrm{ELBO}_{\mathrm{aux}} \geq \mathrm{ELBO}_{\mathrm{init}}$, since they are equal at initialization time and the optimization process does not decrease $\mathrm{ELBO}_{\mathrm{aux}}$. From this combined with our previous result, it follows that $\mathrm{ELBO}_{\mathrm{ref}} \geq \mathrm{ELBO}_{\mathrm{init}}$ and therefore ensuring that the ELBO improves through the refining steps. Note that this also implies that it is not necessary to optimize until convergence: any amount of optimization increases the ELBO.

Figure 2 shows that the ELBO improvement occurs on real world datasets. With the sampling of each auxiliary variable, the ELBO improves forming a *staircase* pattern. Table 1 and 2 serve as further empirical evidence that the ELBO improves as a result of the refinement steps.

**Extending to multiple auxiliary variables**  For simplicity, we stated the arguments with two auxiliary variables, but they straight-forwardly extend to any finite number of auxiliary variables. In this scenario, there are $K$ $\mathrm{ELBO}_{\mathrm{aux}}$-s, one for each auxiliary variable, upper bounded by $\mathrm{ELBO}_{\mathrm{ref}}$ and lower bounded by $\mathrm{ELBO}_{\mathrm{init}}$.

## 3  RELATED WORKS

While in theory, the Bayesian approach can accurately capture uncertainty, in practice we find that exact inference is computationally infeasible in most scenarios and thus we have to resort to approximate inference methods. There is a wealth of research on approximate inference methods, here we focus on works closely related to this paper.

Variational inference (Hinton & Van Camp, 1993) tries to approximate the true posterior distribution over parameters with a variational posterior from a simple family of distributions. Mean-field VI, which for neural networks traces back to Peterson (1987), uses independent Gaussian distributions over the parameters to try to capture the posterior. The advantage of the mean-field approximation is that the network can be efficiently trained using the reparameterization trick (Kingma & Welling, 2013) and the variational posterior has a proper density over the parameter space which then can be used across tasks like continual learning (Osawa et al., 2019; Nguyen et al., 2017) and contextual bandits (Riquelme et al., 2018). Recently, Louizos & Welling (2017) showed that normalizing flows can be employed to further increase the flexibility of the variational posterior. Zhang et al. (2018a) provide a detailed survey of recent advances in VI.

Our method is a novel variant of the auxiliary variable approaches to VI (Agakov & Barber, 2004; Ranganath et al., 2016) that increase the flexibility of the variational posterior through the use of auxiliary variables. The key distinction, however, is that instead of trying to train a complex variational approximation over the joint distribution, we iteratively train simple, mean-field approximations at the sampled values of the auxiliary variables. While this poses an $O(MK)$ ($K$ is the number of auxiliary variables and $M$ is the number of posterior samples) overhead over mean-field VI, the training itself is kept straightforward and efficient. The introduction of every new auxiliary variable increases

the flexibility of the posterior approximation. In contrast to MCMC methods, it is unclear whether the algorithm approaches the true posterior in the limit of infinitely many auxiliary variables.

There are also numerous methods that start with an initial variational approximation and refine it through a few MCMC steps (Salimans et al., 2015; Zhang et al., 2018b; Ruiz & Titsias, 2019). The distinction from our algorithm is that we refine the posterior starting at large scale and iteratively move towards smaller scale refinements whereas these methods only refine the posterior at the scale of the MCMC steps. Guo et al. (2016); Miller et al. (2017) and Locatello et al. (2018) used boosting to refine the variational posterior, iteratively adding parameters such as mixture components to minimize the residual of the ELBO. Our method does not add parameters at training time but instead iteratively refines the samples through the introduction of auxiliary variables. We did not include these models among our baselines because they have yet to be applied to Bayesian multi-layer neural networks.

Further related works include methods that iteratively refine the posterior in latent variable models (Hjelm et al., 2016; Cremer et al., 2018; Kim et al., 2018; Marino et al., 2018). These methods, however, focus on reducing the amortization gap and they do not increase the flexibility of the variational approximation.

Lastly, there are non-Bayesian strategies for estimating epistemic uncertainty in deep learning. Bootstrapping (Breiman, 1996) and deep ensembles (Lakshminarayanan et al., 2017) are perhaps the most promising. Deep ensembles, in particular, have been demonstrated to achieve strong performance on benchmark regression and classification problems and uncertainty benchmarks including out-of-distribution detection (Lakshminarayanan et al., 2017) and prediction under distribution shift (Ovadia et al., 2019). Both methods rely on constructing a set of independently trained models to estimate the uncertainty. Intuitively, the amount of disagreement across models reflects the uncertainty in the ensemble prediction. To induce diversity among the ensemble members, bootstrapping subsamples the training set for each member while deep ensembles use the randomness in weight initialization and mini-batch sampling.

## 4 EXPERIMENTS

The goal of the experiments is twofold. First, we empirically confirm the improvement in the ELBO as claimed in Section 2.3. Second, we quantify the improvement in the uncertainty estimates due to the refinement. We conducted experiments on a selection of regression and classification benchmarks using Bayesian neural networks as the underlying model. We looked at the marginal log-likelihood of the predictions as well as accuracy in classification tasks.

**Refinement (Refined VI)**   After training the initial mean-field approximation, we refine $M = 10$ ensemble members, each with $K = 5$ auxiliary variables. The means on their prior distributions are fixed at 0., and their variances form a geometric series (each auxiliary variable has variance 0.3 times the previous one, which roughly halves the standard deviation of the prior each time): $\sigma_{a_1}^2 = 0.7\sigma_w^2$, $\sigma_{a_2}^2 = 0.21\sigma_w^2$, $\sigma_{a_3}^2 = 0.063\sigma_w^2$, $\sigma_{a_4}^2 = 0.0189\sigma_w^2$, and $\sigma_{a_5}^2 = 0.0081\sigma_w^2$. In each refinement iteration, we optimized the posterior with Adam (Kingma & Ba, 2014) for 200 iterations. To keep the training stable, we scaled the learning rate according to the standard deviation of the conditional posterior: in iteration $k$, lr$=0.3^{\frac{k}{2}}0.001$.

### 4.1 BASELINES

We used three baselines. First, mean-field variational inference in order to quantify the improvement provided by refining. Second, multiplicative normalizing flows (MNF) to compare against a more flexible posterior approximation and third, deep ensemble models to compare against a state-of-the-art non-Bayesian approach. For all methods we used a batch size of 256 and the Adam optimizer with the default learning rate of 0.001[3].

---

[3]The hyperparameters of each baseline were tuned using a Bayesian optimization package, however we found batch size and learning rate to be consistent across methods.

**Variational inference (VI) (Graves, 2011; Blundell et al., 2015)** Naturally, we investigate the improvement over variational inference with a mean-field Gaussian posterior approximation. We do inference over all weights and biases with a Gaussian prior centered at 0. The variance of the prior is tuned through empirical Bayes. This model is trained for 30000 iterations.

**Multiplicative Normalizing Flows (MNF), (Louizos & Welling, 2017)** To measure the performance against a more flexible class of posterior approximations, we look at Multiplicative Normalizing Flows. In this work, the posterior means are augmented with a multiplier from a flexible distribution parameterized by the masked RealNVP. This model is trained with the default flow parameters for 30000 iterations.

**Deep ensemble models, (Lakshminarayanan et al., 2017)** Deep ensemble models are shown to be surprisingly effective at quantifying uncertainty. While they are less principled than Bayesian methods, they are certainly a competitive baseline. For the regression datasets, we used adversarial training ($\epsilon = 0.01$) whereas in classification we did not use adversarial training (adversarial training did not give an improvement on the classification benchmarks). For each dataset, we trained 10 ensemble members for 5000 iterations each.

## 4.2 REGRESSION BENCHMARKS

Following Hernández-Lobato & Adams (2015), we evaluate the methods on a set of UCI regression benchmarks (Table 1). The datasets used a random 80-20 split for training and testing. The model used for these datasets is a feed forward neural network with a single hidden layer containing 50 units with a ReLU activation function. As it is common practice, we utilize the local reparameterization trick (Kingma et al., 2015).

On these benchmarks, we are able to confirm that the refinement step consistently improves both the ELBO and the uncertainty estimates over VI. On 7 out of the 9 datasets, Refined VI is one of the best performing approaches.

| | Deep Ensemble | MNF | VI | | Refined VI (This work) | |
| --- | --- | --- | --- | --- | --- | --- |
| | MLL | MLL | MLL | ELBO | MLL | ELBO |
| boston_housing | -9.136±5.719 | **-2.920**±0.133 | **-2.874**±0.151 | -668.272±7.647 | **-2.851**±0.185 | ≥ -630.379±7.716 |
| concrete_strength | -4.062±0.130 | -3.202±0.055 | **-3.138**±0.063 | -3248.137±68.575 | **-3.131**±0.062 | ≥ -3071.124±64.046 |
| naval_propulsion | 3.995±0.013 | 3.473±0.007 | **5.969**±0.245 | 53440.701±2047.340 | **6.128**±0.171 | ≥ 54882.656±1228.361 |
| energy_efficiency | **-0.666**±0.058 | -0.756±0.054 | -0.749±0.068 | -1296.721±66.310 | **-0.707**±0.094 | ≥ -1192.337±62.089 |
| yacht_hydrodynamics | **-0.984**±0.104 | -1.339±0.170 | -1.749±0.232 | -928.758±112.928 | -1.626±0.231 | ≥ -790.052±84.716 |
| kin8nm | **1.135**±0.012 | **1.125**±0.022 | 1.066±0.019 | 6071.268±61.758 | 1.069±0.018 | ≥ 6172.709±67.659 |
| power_plant | -3.935±0.140 | **-2.835**±0.033 | **-2.826**±0.020 | -22496.579±130.487 | **-2.820**±0.024 | ≥ -22368.965±85.308 |
| protein_structure | -3.687±0.013 | **-2.928**±0.007 | **-2.926**±0.010 | -108806.007±174.522 | **-2.923**±0.009 | ≥ -108597.593±158.482 |
| wine | **-0.968**±0.079 | **-0.963**±0.027 | **-0.973**±0.054 | -1346.130±18.004 | **-0.968**±0.056 | ≥ -1311.898±17.487 |

Table 1: Results on the UCI regression benchmarks with a single hidden layer containing 50 units. Metrics: marginal log-likelihood (MLL, higher is better), and the evidence lower bound (ELBO higher is better). The mean values and standard deviations are shown in the table.

## 4.3 CLASSIFICATION BENCHMARKS

We examine the performance on commonly used image classification benchmarks (Table 2). The architecture used for this experiment is the LeNet5 (LeCun et al., 1995) architecture containing three convolutional layers with 6, 16 and 120 channels respectively followed by a feed-forward layer with 84 units and an output layer with ReLu activations throughout the network. We use the local reparameterization trick (Kingma et al., 2015) for the dense layers and Flipout (Wen et al., 2018) for the convolutional layers to reduce the gradient noise.

On the classification benchmarks, we again are able to confirm that the refinement step consistently improves both the ELBO and the uncertainty estimates over VI. While Refined VI is unable to outperform Deep Ensembles in classification accuracy, it does outperform them in MLL on the largest dataset, CIFAR10.

| | Deep Ensemble | MNF | VI | | Refined VI (This work) | |
|---|---|---|---|---|---|---|
| | MLL & Acc | MLL & Acc | MLL & Acc | ELBO | MLL & Acc | ELBO |
| mnist | **-0.017**±0.001 99.4%±0.0 | -0.034±0.002 99.1%±0.1 | -0.032±0.001 99.1%±0.1 | -7618.533±47.589 | -0.025±0.001 99.2%±0.0 | ≥ -6310.824±42.357 |
| fashion_mnist | **-0.201**±0.002 93.1%±0.1 | -0.255±0.004 90.7%±0.2 | -0.255±0.003 90.7%±0.1 | -22830.330±232.654 | -0.241±0.004 91.3%±0.2 | ≥ -20438.955±79.672 |
| cifar10 | -0.791±0.009 76.3%±0.3 | -0.795±0.013 72.8%±0.6 | -0.815±0.004 72.3%±0.5 | -57257.887±299.570 | **-0.768**±0.007 73.5%±0.5 | ≥ -50989.217±238.976 |

Table 2: Results on image classification benchmarks with the LeNet-5 architecture, without data augmentation. Metrics: marginal log-likelihood (MLL, higher is better), accuracy (Acc, higher is better), and the evidence lower bound (ELBO higher is better). The mean values and standard deviations are shown in the table.

## 4.4 LARGE SCALE MODELS

To demonstrate the performance on larger scale models, we apply the refining algorithm to Residual Networks (He et al., 2016) with 20 layers (based on Keras's ResNet implementation). We look at two models. Firstly, a model where we do inference over all of the residual blocks and secondly, following Ovadia et al. (2019), a hybrid model (ResNet Hybrid) where inference is only done over the final layer of each residual block and every other layer is treated as a regular layer. For this model, we used a batch-size of 256 and we decayed the learning rate starting from 0.001 over 200 epochs. We used 10 auxiliary variables each reducing the prior variance by a factor of 0.5. Furthermore we investigate the effect of Batch Normalization (Ioffe & Szegedy, 2015). While it is difficult to incorporate batch normalization into the Bayesian framework, its positive effects are undeniable.

Regarding batch normalization, we can confirm the findings of Osawa et al. (2019), that it provides a substantial improvement for VI, although interestingly, this improvement disappears in the case of the hybrid model. To our knowledge, the refined hybrid model is state-of-the-art in terms of MLL. It outperforms Osawa et al. (2019) in both MLL and accuracy.

| | Deep Ensemble | | VI | | Refined VI (This work) | |
|---|---|---|---|---|---|---|
| | MLL | Acc | MLL | Acc | MLL | Acc |
| ResNet | -0.698 | 82.7% | -0.795 | 72.6% | **-0.696** | 75.5% |
| ResNet + BatchNorm | **-0.561** | 83.6% | -0.672 | 77.6% | -0.593 | 79.7% |
| ResNet Hybrid | -0.698 | 82.7% | -0.465 | 84.2% | **-0.432** | 85.8% |
| ResNet Hybrid + BatchNorm | -0.561 | 83.6% | -0.465 | 84.0% | **-0.423** | 85.6% |

Table 3: Results on CIFAR10 with the ResNet architecture, without data augmentation. Metrics: marginal log-likelihood (MLL, higher is better), accuracy (Acc, higher is better), and the evidence lower bound (ELBO higher is better). Note that the non-hybrid and the hybrid models are equivalent when trained deterministically.

## 4.5 COMPUTATIONAL COSTS

When introducing a novel algorithm for variational inference, we have to discuss the computational costs. Table 4 shows the wall-time required to train each model including the possibility of parallel training. Deep ensembles require fewer training iterations and parallelize very well, although it is important to note that the system we tested on is heavily optimized towards training these models both in hardware and software. For Refined VI, training the initial mean-field approximation cannot be parallelized, but the following refinement iterations can be straight-forwardly split into $M$ threads.

| | No parallelism | Maximum parallelism |
|---|---|---|
| Deep Ensemble | 433.7 s | 43.4 s |
| MNF | 990.9 s | 990.9 s |
| VI | 531.5 s | 531.5 s |
| Refined VI | 708.6 s | 566.9 s |

Table 4: The training time of each method (LeNet-5/CIFAR10) on a P100 using Tensorflow.

## 4.6 CONCLUSIONS

In this paper, we describe a novel algorithm for refining a coarse variational approximation to the Bayesian posterior. We show, both theoretically and empirically, that the refined posterior is a better approximation to the posterior than the initial variational distribution. Our method outperforms the baseline variational approximations in both uncertainty estimation as well as computational requirements. It sets a new state-of-the-art in uncertainty estimation using variational inference at ResNet scale (ResNet-20) on CIFAR10.

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

## A CLOSED FORMS OF THE SAMPLING DISTRIBUTIONS

In Section 2.2, we claim that $q_\phi(a_1)$ (used to sample $a_1$) and $q_\phi(w|a_1)$ (used to initialize $q_{\phi_1}$) have analytic solutions for additive Gaussians. For $p(w) = \mathcal{N}(0, \sigma_w^2)$, $q(w) = \mathcal{N}(\mu_\phi, \sigma_\phi^2)$, $p(a_1) = \mathcal{N}(0, \sigma_{a_1}^2)$ and $p(a_2) = \mathcal{N}(0, \sigma_{a_2}^2)$,

$$q_\phi(a_1) = \int p(a_1|w) q_\phi(w)\, \mathrm{d}w = \mathcal{N}\Big(\mu_\phi \frac{\sigma_{a_1}^2}{\sigma_w^2}, \frac{\sigma_\phi^2 \sigma_{a_1}^4}{\sigma_w^4} + \frac{\sigma_{a_1}^2 \sigma_{a_2}^2}{\sigma_w^2}\Big)$$

$$q_{\phi_1}(w) = \mathcal{N}\Big(\mu_{\phi_1}, \sigma_{\phi_1}^2\Big) \leftarrow q_\phi(w|a_1) = \mathcal{N}\Big(\frac{a_1 \sigma_\phi^2 \sigma_w^2 + \mu_\phi \sigma_{a_2}^2 \sigma_w^2}{\sigma_\phi^2 \sigma_{a_1}^2 + \sigma_w^2 \sigma_{a_2}^2}, \frac{\sigma_\phi^2 \sigma_w^2 \sigma_{a_2}^2}{\sigma_{a_1}^2 \sigma_\phi^2 + \sigma_w^2 \sigma_{a_2}^2}\Big).$$

(1)

These are derived by applying Bayes rule to the Gaussian probability density functions.

## B NOTE ON $\mathrm{ELBO}_{\mathrm{AUX}} \geq \mathrm{ELBO}_{\mathrm{INIT}}$

We formally show that $\mathrm{ELBO}_{\mathrm{aux}} \geq \mathrm{ELBO}_{\mathrm{init}}$ under the assumption that the conditional variational posterior, $q_\phi(w|a_1)$, is within the variational family of $q_{\phi_1}$ for all values of $a_1$.

For a given $a_1$, let $\phi_{1_0|a_1}$ be a set of variational parameters such that $\forall w, q_{\phi_{1_0|a_1}}(w) = q_\phi(w|a_1)$. Such $\phi_{1_0|a_1}$ must exist as a result of the initial assumption. The formula for Gaussian distributions is given in Appendix A.

Set $\phi_{1|a_1}$[4] such that

$$\phi_{1|a_1} = \begin{cases} \phi_{1_0|a_1} & \text{if } \mathbb{E}_{q_{\phi_{1_0|a_1}}}\Big[\log p(y|x,w) - \log \frac{q_{\phi_{1_0|a_1}}(w)}{p(w|a_1)}\Big] \geq \mathbb{E}_{q_{\phi_{1_{opt}|a_1}}}\Big[\log p(y|x,w) - \log \frac{q_{\phi_{1_{opt}|a_1}}(w)}{p(w|a_1)}\Big] \\ \phi_{1_{opt}|a_1} & \text{otherwise} \end{cases}$$

(2)

where $\phi_{1_{opt}|a_1}$ is the result of the stochastic optimizer that attempts to maximize $\mathbb{E}_{q_{\phi_{1|a_1}}}\Big[\log p(y|x,w) - \log \frac{q_{\phi_{1|a_1}}(w)}{p(w|a_1)}\Big]$.

By construction, for all $a_1$,

$$\mathbb{E}_{q_{\phi_{1|a_1}}}\Big[\log p(y|x,w) - \log \frac{q_{\phi_{1|a_1}}(w)}{p(w|a_1)}\Big] \geq$$

$$\mathbb{E}_{q_{\phi_{1_0|a_1}}}\Big[\log p(y|x,w) - \log \frac{q_{\phi_{1_0|a_1}}(w)}{p(w|a_1)}\Big] =$$

$$\mathbb{E}_{q_\phi(w|a_1)}\Big[\log p(y|x,w) - \log \frac{q_\phi(w|a_1)}{p(w|a_1)}\Big].$$

(3)

Substituting this into $\mathrm{ELBO}_{\mathrm{aux}}$ gives

$$\mathrm{ELBO}_{\mathrm{aux}} =$$

$$\mathbb{E}_{q_\phi}\Big[\mathbb{E}_{q_{\phi_{1|a_1}}}\Big[\log p(y|x,w) - \log \frac{q_{\phi_{1|a_1}}(w)}{p(w|a_1)}\Big] - \log \frac{q_\phi(a_1)}{p(a_1)}\Big] \geq$$

$$\mathbb{E}_{q_\phi}\Big[\mathbb{E}_{q_\phi(w|a_1)}\Big[\log p(y|x,w) - \log \frac{q_\phi(w|a_1)}{p(w|a_1)}\Big] - \log \frac{q_\phi(a_1)}{p(a_1)}\Big] =$$

$$\mathbb{E}_{q_\phi}\Big[\log p(y|x,w) - \log \frac{q_\phi(w)}{p(w)}\Big] =$$

$$\mathrm{ELBO}_{\mathrm{init}}$$

(4)

concluding the proof.

---

[4]We use the notation $\phi_{1|a_1}$ instead of $\phi_1$ to emphasize the dependence on the value of $a_1$.

