# OpenReview forum: "Refining the variational posterior through iterative optimization"
_ICLR.cc/2020/Conference — Reject_

### Official Review · AnonReviewer3 · 2019-10-23
**Official Blind Review #3**

**Rating:** 6

**Review:**

Summary.

This paper describes a method for training flexible variational posterior distributions, which consists in making iterative locale refinements to an initial mean-field approximation, using auxiliary variables. The focus is on Gaussian latent variables, and the method is applied to Bayesian neural nets to perform variational inference (VI) over the weights. Empirical results show improvements upon the performance of the mean-field approach and some other baselines, on classification and regression tasks.

Main comments.

Overall, this paper is well written and easy to follow. It tackles an important topic in VI and proposes an interesting idea to improve the flexibility of the approximate distribution. I have the following comments/questions.

- On the guarantee of improvement. I still have some doubts regarding the inequality “ELBO_aux >=  ELBO_init”. Can you please elaborate more on this and provide a detailed formal proof? Figure 2 shows that ELBO_aux can go below ELBO_init.
- The focus of the paper is on Gaussian variables and a configuration where some key distributions, q(a_1) and q(w|a_1), are accessible in closed from. The generalization of the proposed method beyond these settings should be discussed and explored in experiments.
- Important baselines are missing in the experiments. I would recommend including at least the other VI techniques relying on auxiliary variables to build flexible variational families [1,2]. This would help to better assess the impact/importance of the proposed method.

[1] Ranganath, Rajesh, Dustin Tran, and David Blei. "Hierarchical variational models." ICML. (2016).
[2] Maaløe, Lars, et al. "Auxiliary deep generative models." ICML (2016).


**Experience Assessment:**

I have read many papers in this area.

**Review Assessment: Checking Correctness Of Derivations And Theory:**

I assessed the sensibility of the derivations and theory.

**Review Assessment: Checking Correctness Of Experiments:**

I assessed the sensibility of the experiments.

**Review Assessment: Thoroughness In Paper Reading:**

I read the paper thoroughly.

---

> ### Author Response · Authors · 2019-11-12
> **Reply to Review #3**
>
> Thank you for the review and feedback. We appreciate that you found the paper interesting.
>
> 1. The argument is that there exists a phi_1, for which ELBO_aux = ELBO_init. Therefore, by further optimizing phi_1, we can ensure that ELBO_aux >= ELBO_init. If the optimizer were to produce a phi_1 for which ELBO_aux < ELBO_init (since it is a stochastic optimizer), we can simply discard the result of the optimizer and proceed with the value of phi_1 for which ELBO_aux = ELBO_init. This guarantees that ELBO_aux >= ELBO_init.
>
> To further clarify the claim, we are going to include a formal proof in the appendix stating that max(ELBO_aux(phi_1), ELBO_aux(optimized(phi_1))) >= ELBO_aux(phi_1) = ELBO_init.
>
> The phi_1 where ELBO_aux = ELBO_init can be analytically computed (formula given in the Appendix) by ensuring that q_phi(w|a1)=q_{phi_1}(w). In the experiments, we use this formula to initialise phi_1 for the SGD training.
>
> Regarding Figure 2, the drops are due to optimiser artefacts. When we initialise phi_1, the momentum term of Adam is reset and it takes a few iterations to find a good local optima. If we used a smaller learning rate these drops would not occur, but convergence would be slower.
>
> 2. One of the benefits of the approach is that it is very general. It can be applied to any probabilistic model with any joint distribution p(w, a1, …, aK) (obeying the constraints specified in the paper), meaning that the auxiliary variables do not have to be additive or independent, they can have arbitrary distributions. We showcased the method in Bayesian neural networks, because they have a challenging posterior distribution. Application to variational autoencoders, latent variable models etc. with different variational distributions is certainly interesting and we are considering it for future work.
>
> The existence of the analytical conditionals is only used once in the paper when we show the guarantee of improvement (ELBO_aux >= ELBO_init). In the case when the conditional posterior is not analytically computable, this guarantee does not hold, but the algorithm can still be applied and it might provide an improvement.
>
> 3. [1,2] are indeed relevant and interesting papers and we discuss them in the related works section. However, an open challenge is how to apply auxiliary variables for Bayesian neural networks, which have a significantly large parameter space . (HVM is applied to Deep exponential families and Auxiliary deep generative models is used for generative models where posterior dimensions are <1000.) Multiplicative Normalizing Flows (Louizos and Welling, ‎2017) builds on those papers for Bayesian neural networks and is a method we compare to. Other SOTA is represented by Deep Ensembles (Lakshminarayanan et al., 2017) and VOGN (Osawa et al., ‎2019).
>
>
> (Lakshminarayanan et al., 2017) Lakshminarayanan, Balaji, Alexander Pritzel, and Charles Blundell. "Simple and scalable predictive uncertainty estimation using deep ensembles." Advances in Neural Information Processing Systems. 2017.
>
> (Osawa et al., ‎2019) Osawa, Kazuki, et al. "Practical Deep Learning with Bayesian Principles." Advances in Neural Information Processing Systems. 2019.
>
> (Louizos and Welling, ‎2017) Louizos, Christos, and Max Welling. "Multiplicative normalizing flows for variational Bayesian neural networks." Proceedings of the 34th International Conference on Machine Learning-Volume 70. JMLR. org, 2017.

---

> > ### Comment · AnonReviewer3 · 2019-11-15
> > **Comments on the rebuttal**
> >
> > Thanks for your response.
> >
> > 1. On the guarantee of improvement. Thanks for adding the formal proof, which is helpful and addresses my doubts. The discussion on the improvement over ELBO_init is still a bit hard to follow, due to the different notations involved and the order in which things are presented. I would therefore suggest to refactor this part as follows: start by stating that ELBO_aux = ELBO_init if q_phi1(w) = q_phi(w|a_1), then ELBO_aux >= ELBO_init would follow thanks to optimizing ELBO_aux over phi_1, and finally discuss the necessary conditions to initialize q_phi1(w) with q_phi(w|a_1).
> >
> > 2. On the generality. The paper only treats the case where the conditional posterior is accessible analytically, and it would benefit from discussions/analysis of situations where q(w|a1) is implicit, i.e., we cannot evaluate its density. Ideally, experiments under the latter setting should also be included, especially since the inequality ELBO_aux >= ELBO_init may no longer hold.  Otherwise, it would be hard to convince one to consider the proposed method beyond settings where the conditional posterior is accessible.
> >
> > 3. On missing baselines. HVM, for instance, can be reasonably considered in the experiments of sections 4.2 and 4.3. As you mentioned it has been applied to Deep Exponential Families (DEFs), which generalizes Bayesian feedforward neural nets by using stochastic layers. In other words, under comparable architectures, inference is even more challenging with DEFs than Bayesian neural nets. Moreover, I could not find empirical comparisons with HVM or Auxiliary Deep Generative models in (Louizos and Welling, ‎2017).

---

### Official Review · AnonReviewer1 · 2019-10-24
**Official Blind Review #1**

**Rating:** 6

**Review:**

The paper proposes a new way to improving the variational posterior. The q(w) is an integral over multiple auxiliary variables. These auxiliary variables can be specified at different scales that can be refined to match different scales of details of the true posterior. They show better performance regression and classification benchmark datasets. They also show that the training time is at a reasonable scale when being parallelized.

I think the idea is quite interesting. They did a good illustration of the difference between their model and related works. They also compare with the state-of-art variational approximation methods.

One concern I have is the complexity. I don't think it's just O(MK) since it has to optimize for each phi_k for each posterior sample w. This could be quite large depending on problems.

Also is the refining only run for once or run after each update of the mean field q(w)? If it's the latter, the overhead would be much larger.

When w is very high-dimensional, the number of auxiliary variables should be exponentially larger. Is that true? Or it's actually invariant to the dimensionality of the posterior distribution?

The paper proves that the refined ELBO is larger than the auxiliary ELBO which is larger than the initial mean field. But the initial ELBO should be a tight lower bound of the true log likelihood. Would that be a problem that ELBO_ref actually spill over the true log likelihood which is a dangerous sign?

Overall I think the paper is well written. The experiments are carefully designed. The idea is interesting and useful.

**Experience Assessment:**

I have published one or two papers in this area.

**Review Assessment: Checking Correctness Of Derivations And Theory:**

I assessed the sensibility of the derivations and theory.

**Review Assessment: Checking Correctness Of Experiments:**

I assessed the sensibility of the experiments.

**Review Assessment: Thoroughness In Paper Reading:**

I read the paper at least twice and used my best judgement in assessing the paper.

---

> ### Author Response · Authors · 2019-11-12
> **Reply to Review #1**
>
> Thank you for the review and feedback. We appreciate that you found the paper interesting.
>
> Regarding the complexity, it is indeed the case that phi_k needs to be optimised for each sample w. There are O(MK) refinement steps, where each refinement step amounts to 200 steps of stochastic gradient descent which represents a ~25% computational overhead. For empirical numbers, see Section 4.5.
>
> The initial mean-field q(w) is trained before any refinement step takes place and it is not changed after the refinement steps.
>
> The dimensionality of a is the same as the dimensionality of w. Any number of a1, ..., ak can be used for the refinement, it is invariant of the dimensionality of w. In our experiments we used a1, …, a5 because each new auxiliary variable comes with further computational overhead.
>
> It is true that the ELBO_init is a lower bound to the marginal likelihood but, this lower bound is only tight when the initial q(w) is the true posterior. In this case, the refinement steps would not provide any improvement, resulting in log p(y|x)=ELBO_ref=ELBO_aux=ELBO_init.
>
> log p(y|x) >= ELBO_ref holds, because the refined VI is still a variational inference approach, so the ELBO of the refined distribution is still a lower bound to the marginal likelihood.

---

### Official Review · AnonReviewer4 · 2019-11-01
**Official Blind Review #4**

**Rating:** 6

**Review:**

The paper proposes to improve standard variational inference by increasing the flexibility of the variational posterior by introducing a finite set of auxiliary variables. Motivated by the limited expressivity of mean field variational inference the author suggests to iteratively refine a ‘proposal’ posterior by conditioning on a sequence of auxiliary variables with decreasing variance. The key requirement to set the variance of the auxiliary variables such that the integrating over them leaves the original model unchanged. As noted by the authors this is a variant of auxiliary variables introduced by Barber & Agakov. The motivation and theoretical sections seems sound and the experimental results are encouraging, however maybe not completely supporting the claim of new ‘state of the art’ on uncertainty prediction.

Overall i find the motivation and theoretical contribution interesting. However I do not find the experimental section completely comprehensive why I currently think the paper is borderline acceptance.

Comments
1) The computational demand using the method seems quite large by adding O(NumSamples * NumAuxiliary) additional computational cost on top of the standard VI. Here each factor M is quite large e.g. 200 epochs for CIFAR10 (if i understand the algorithm correctly?)
2) For the UCI experiments the comparison is only made against DeepEnsembels or other VI methods, however to the best of my knowledge MCMC methods are superior in this setting given the small dataset size?
3) The results on CIFAR10 do seem to demonstrate that the proposed method is superior to DeepEnsembles and standard VI in one particular setting where VI is only performed over a small subset of layers in a ResNet (why doesn’t it work for when doing VI on all the parameters?). However generally looking at the best obtained results of ~86% acc this is quite far from current best probabilistic models (see e.g. Heek2019 that gets 94% acc). Some of this can probably be attributed to differences in data-augmentation and model architecture however in general it makes it very hard to compare with other methods when the baselines are not competitive.

Minor Comments:
 In relation to comment 3) above I think you should reword the sentence “It sets a new state-of-the-art in uncertainty estimation at ResNet scale on CIFAR10” in the conclusion.


“In  order  to  get  independent  samples  from  the  variational  posterior,we have to repeat the iterative refinement for each ensemble member”: Does this imply that if we want M samples we first have to optimize using the standard VI and then to M optimizations to get q_k(w)?


How sensitive is the method to sequence of variances for a?

[Heek2019]: Bayesian Inference for Large Scale Image Classification


**Experience Assessment:**

I have published one or two papers in this area.

**Review Assessment: Checking Correctness Of Derivations And Theory:**

I assessed the sensibility of the derivations and theory.

**Review Assessment: Checking Correctness Of Experiments:**

I assessed the sensibility of the experiments.

**Review Assessment: Thoroughness In Paper Reading:**

I read the paper at least twice and used my best judgement in assessing the paper.

---

> ### Author Response · Authors · 2019-11-12
> **Reply to Review #4**
>
> Thank you for the review and feedback. We appreciate that you found the paper interesting.
>
> 1. The refinement steps can be computationally demanding, but any amount of refinement is guaranteed to improve the approximate posterior. In our experiments, the computational cost of the refinement steps is roughly 25% of the cost of training the initial mean-field approximation, which represents a non-trivial, but feasible computational overhead. For the LeNet-5 experiments, the refinement steps amount to 50 epochs, while in the ResNet experiments, we have more relaxed computational constraints so the refinement steps add up to 200 epochs. See Section 4.5 for cost comparisons, where the SOTA in expressive posteriors (multiplicative normalizing flows) is more compute-expensive.
>
> 2. MCMC methods can show strong performance on small scale regression tasks, and given enough time, they might converge to a better posterior approximation (May we ask for a citation for exact numbers on the UCI benchmarks?). Gaussian processes and deep Gaussian processes (Salimbeni and Deisenroth, 2017) are also known to be competitive on these benchmarks. In our paper, the regression experiments are meant to serve as a comparison to other variational approaches and we do not claim SOTA performance.
>
> 3. We believe that the core contribution of our paper is an interesting and original approach to VI, rather than the specific SOTA results.
>
> The baselines that we compare against at ResNet scale are Deep Ensembles (Lakshminarayanan et al., 2017), Variational Inference (Ovadia et al., 2019) and Variational Gauss-Newton  (Osawa et al., ‎2019) which can be considered state-of-the art. Refined VI outperforms these on ResNet20. We agree that our wording, “It sets a new state-of-the-art in uncertainty estimation at ResNet scale on CIFAR10” suggests a more general result and we are going to adjust the phrasing to reflect the model size and the setting. We are changing the phrasing of our introduction and conclusion sections to put less emphasis on the specific results and focus on the benefits of the core idea instead.
>
> (Heek and Kalchbrenner, 2019) shows very strong results, however, as the review also mentions, a direct comparison is problematic, since they use a significantly larger model (ResNet56), proposed changes to the architecture, and use significantly more compute (1000 epochs). They are also a concurrent ICLR submission. Nevertheless, we are impressed by the results and we are eager to see this paper being published soon.
>
> 4. The refinement steps have to be repeated for each ensemble member meaning that with K auxiliary variables and M ensemble members, one has to refine MK times after training the initial mean-field approximation. This amounts to a ~25% computational overhead compared to standard VI.
>
> 5. The method is not sensitive to the variances of a1,..,a5. We set their variances so that they form a decreasing geometric sequence with factor 0.7, but any factor between 0.3 and 0.9 performed similarly.
>
> If this reply addressed your main comments, please consider revising your score, otherwise let us know the remaining concerns you might have.
>
> (Salimbeni and Deisenroth, 2017) Salimbeni, Hugh, and Marc Deisenroth. "Doubly stochastic variational inference for deep Gaussian processes." Advances in Neural Information Processing Systems. 2017.
>
> (Lakshminarayanan et al., 2017) Lakshminarayanan, Balaji, Alexander Pritzel, and Charles Blundell. "Simple and scalable predictive uncertainty estimation using deep ensembles." Advances in Neural Information Processing Systems. 2017.
>
> (Ovadia et al., 2019) Ovadia, Yaniv, et al. "Can You Trust Your Model's Uncertainty? Evaluating Predictive Uncertainty Under Dataset Shift." Advances in Neural Information Processing Systems. 2019.
>
> (Osawa et al., ‎2019) Osawa, Kazuki, et al. "Practical Deep Learning with Bayesian Principles." Advances in Neural Information Processing Systems. 2019.
>
> (Heek and Kalchbrenner, 2019) Heek, Jonathan, and Nal Kalchbrenner. "Bayesian Inference for Large Scale Image Classification." arXiv preprint arXiv:1908.03491 (2019).

---

> > ### Comment · AnonReviewer4 · 2019-11-14
> > **Reponse 2**
> >
> > Thanks for the answers
> >
> >
> > "MCMC methods can show strong performance on small scale regression tasks, and given enough time, they might converge to a better posterior approximation (May we ask for a citation for exact numbers on the UCI benchmarks?)"
> >
> > I trawled google scholar for a refence but actually couldn't find anything useful - I'll rest my case on this one then :)
> >
> > The refinement steps have to be repeated for each ensemble member meaning that with K auxiliary variables and M ensemble members, one has to refine MK times after training the initial mean-field approximation. This amounts to a ~25% computational overhead compared to standard VI.
> >
> > Can you clarify how you arrive and 25% overhead here e.g. what type of standard VI are you comparing against?
> >
> >
> > "(Heek and Kalchbrenner, 2019) shows very strong results, however, as the review also mentions, a direct comparison is problematic, since they use a significantly larger model (ResNet56), proposed changes to the architecture, and use significantly more compute (1000 epochs). They are also a concurrent ICLR submission. Nevertheless, we are impressed by the results and we are eager to see this paper being published soon."
> >
> > Yes some can definitely be attributed to model differences. I know it is a lot of work but I think it would make your paper a lot stronger if you could show encouraging results on a reasonably competitive model e.g. ResNet56 on Cifar10? Otherwise I think rephrasing your text a bit to be less focused on SOTA results and more on the conceptual contribution is also a reasonable option as you suggests your self.

---

### Official Review · AnonReviewer2 · 2019-11-06
**Official Blind Review #2**

**Rating:** 3

**Review:**

This paper presented an iteratively refined variational inference for Gaussian latent variables. The intuition is straightforward and makes sense to me. However, I have some concerns.

Detailed comments:
1. In theoretical justification, only K=2 is discussed. My intuition is that as K increases, the approximation of the true posterior should be closer. The summation of multiple Gaussian distributions can arbitrarily approximate any distribution given enough base distributions. I would like to see some theoretical discussion about K. At least in the experiment, the author should provide the performance of different Ks.
2. The toy example in the paper is simply 1D Gaussian. I want to see more discussion for high dimensional latent variables. So in the experiments, how you parameterized the distribution for each weight? Totally independent? or allowing structural correlations? I am not sure the details of the implementation in this paper, but I also have a naive question for high dimensional Gaussian. Does it require to compute the matrix inverse when sampling a_k?
3. Another related paper "Guo, Fangjian, et al. "Boosting variational inference." arXiv preprint arXiv:1611.05559 (2016)." should be discussed as well.

**Experience Assessment:**

I have published one or two papers in this area.

**Review Assessment: Checking Correctness Of Derivations And Theory:**

I assessed the sensibility of the derivations and theory.

**Review Assessment: Checking Correctness Of Experiments:**

I assessed the sensibility of the experiments.

**Review Assessment: Thoroughness In Paper Reading:**

I read the paper at least twice and used my best judgement in assessing the paper.

---

> ### Author Response · Authors · 2019-11-12
> **Reply to Review #2**
>
> Thank you for the review and feedback.
>
> 1. We share your intuition that at the limit of infinitely many auxiliary variables, the variational posterior can be very flexible and might approximate the exact posterior exactly in some cases but we were unable to prove this. The strongest statement we can state is that each refinement step is guaranteed to make the refined posterior better. Figure 2 depicts how the ELBO improves with the introduction of each new auxiliary variable (K=1..5).
>
> 2. In our experiments, we train independent Gaussians for all weights (for both the initial and refined posteriors) parameterized by their means and variances. The dimensionality of w, a1, a2, …, ak is the same as the number of weights. The correlations and multi-modality are introduced through the refinement steps as demonstrated in the toy example. The method requires no full matrix inversion (only inversion of diagonal matrices).
>
> 3. Thank you for bringing our attention to this related paper. We are updating the paper to include a brief discussion on it.
>
> In this work, we propose an algorithm for refining the variational posterior of a Bayesian neural network. The refinement enables the variational posterior to capture complex, multi-modal distributions at the cost of a small computational overhead. We present theoretical guarantees as well as empirical evidence that the method provides a significant improvement over standard VI.
>
> If this reply addressed your main comments, please consider revising your score, otherwise let us know the remaining concerns you might have.

---

### Author Response · Authors · 2019-10-25
**Typo in section 2.2**

In Section 2.2, we repeatedly referred to the iteration number with letter 'i' instead of 'k'. We apologise for the confusion that this typo may have caused. We are going to correct it as soon as possible.

---

### Author Response · Authors · 2019-11-12
**Revision**

We appreciate all the reviews and feedback. We made the following changes to address the concerns expressed in the reviews:

Review #2: Brief discussion of (Guo et al., 2016) ‘Boosting variational inference’ in the related works section.

Review #3: Included a formal proof of the claim that ELBO_aux >= ELBO_init in the appendix.

Review #4: Rephrased the misleading claim ‘It sets a new state-of-the-art in uncertainty estimation at ResNet scale on CIFAR10’ as suggested.

Minor:

Fixed the notation of the iteration indices in Section 2.2.

---

### Decision · Program_Chairs · 2019-12-19

**Decision:**

Reject

**Comment:**

In this paper a method for refining the variational approximation is proposed.

The reviewers liked the contribution but a number reservations such as missing reference made the paper drop below the acceptance threshold. The authors are encouraged to modify paper and send to next conference.

Reject.